Virtual reconstruction of the endocranial anatomy of the early Jurassic marine crocodylomorph Pelagosaurus typus (Thalattosuchia)

Pierce Stephanie E. spierce@oeb.harvard.edu 1
Williams Megan 2
Benson Roger B.J. 3
1 Museum of Comparative Zoology and Department of Organismic and Evolutionary Biology, Harvard University , Cambridge , MA , United States
2 Department of Earth Sciences, University of Cambridge , Cambridge , UK
3 Department of Earth Sciences, University of Oxford , Oxford , UK
Knoll Fabien
Electronic publication date: 2017 Apr 25
Publication date: 2017
Volume: 5
Electronic Location ID: e3225
Received 2016 Oct 19; Accepted 2017 Mar 24
Copyright: ©2017 Pierce et al.
Copyright year: 2017
Copyright holder: Pierce et al.
License: This is an open access article distributed under the terms of the Creative Commons Attribution License, which permits unrestricted use, distribution, reproduction and adaptation in any medium and for any purpose provided that it is properly attributed. For attribution, the original author(s), title, publication source (PeerJ) and either DOI or URL of the article must be cited.
License URL: https://creativecommons.org/licenses/by/4.0/

Keywords: Thalattosuchia, Three-dimensional, Marine, Crocodylomorpha, Archosauria, Labyrinth, Sensory adaptation, Endocast, Gharial

Funding: The authors received no funding for this work.

==============================
Thalattosuchians were highly specialised aquatic archosaurs of the Jurassic and Early Cretaceous, and represent a peak of aquatic adaptation among crocodylomorphs. Relatively little is known of their endocranial anatomy or its relevance for the evolution of sensory systems, physiology, and other aspects of biology. Nevertheless, such data have significance for two reasons: (1) thalattosuchians represent an important data point regarding adaptation to marine life in tetrapods; and (2) as early-diverging members of the crocodylian stem-lineage, thalattosuchians provide information on the evolutionary assembly of the brain and other endocranial structures in crocodylomorphs. Here we use µCT data to virtually reconstruct the endocranial anatomy of Pelagosaurus typus, an early thalattosuchian with plesiomorphic traits of relevance to the split between the two major subgroups: Teleosauroidea and Metriorhynchoidea. Interpretation of these data in a broad comparative context indicate that several key endocranial features may be unique to thalattosuchians, including: a pyramidal morphology of the semicircular canals, the presence of an elongate endosseous cochlear duct that may indicate enhanced hearing ability, the presence of large, paired canals extending anteriorly from an enlarged pituitary fossa, a relatively straight brain (possibly due to the presence of large, laterally placed orbits), and an enlarged venous sinus projecting dorsally from the endocast that is confluent with the paratympanic sinus system. Notably, we document a large expansion of the nasal cavity anterior to the orbits in Pelagosaurus as an osteological correlate of an enlarged salt gland previously only documented in Late Jurassic metriorhynchoids. This is the first anatomical evidence of this structure in early thalattosuchians. Pelagosaurus also shares the presence of paired olfactory bulbs with metriorhynchoids, and shows an enlarged cerebrum, which may also be present in teleosauroids. Taken together, our findings indicate that physiological and sensory adaptations to marine life occurred early in thalattosuchian evolution, predating the origins of flippers, tail flukes, and hydrodynamic body forms seen later in metriorhynchoids.

Introduction

Thalattosuchia is a radiation of aquatic crocodylomorphs that attained a worldwide distribution at low latitudes during the Early Jurassic—Early Cretaceous (Mannion et al., 2015). They are characterized by having a longirostrine skull morphology (long, narrow snout), although some taxa possessed relatively shorter and more robust snouts (e.g., Dakosaurus; Gasparini, Pol & Spalletti, 2006). The group is divided into two major clades, the teleosauroids, which were ‘gavial-like’ near shore predators, and the highly-derived pelagic metriorhynchoids (i.e., metriorhynchids), which exhibited modified flipper-like forelimbs, a crescentic fish-like tail, and loss of dermal armour. The morphology, phylogeny, and evolutionary dynamics of the Thalattosuchia has been under intense investigation over the past decade (e.g., Mueller-Töwe, 2005; Mueller-Töwe, 2006; Jouve, 2009; Pierce, Angielczyk & Rayfield, 2009a; Pol & Gasparini, 2009; Young & De Andrade, 2009; Young et al., 2010; Young, Bell & Brusatte, 2011; Martin & Vincent, 2013; Stubbs et al., 2013; Cau, 2014; Martin et al., 2014; Jouve et al., 2016; Young et al., 2016) with some recent studies suggesting that thalattosuchians may be relatively early diverging members of the Crocodyliformes (e.g., Wilberg, 2015a). Moreover, there has been increasing interest in reconstructing their functional paleoecology, with studies focusing on feeding mechanics and niche partitioning (e.g., Pierce, Angielczyk & Rayfield, 2009b; De Andrade et al., 2010; Young et al., 2010; Stubbs et al., 2013; Young et al., 2013), as well as adaptations for aquatic locomotion (e.g., Hua, 1994; Hua & De Buffrenil, 1996; Hua, 2003; Molnar et al., 2015).

A further area of research has concentrated on thalattosuchian endocranial anatomy. Seeley (1880) provided one of the earliest neuroanatomical descriptions of a thalattosuchian by longitudinally sectioning a teleosauroid braincase from the Whitby Lias (Early Jurassic of England); he noted that the brain “differed remarkably from that of living Crocodiles (p. 629)” and commented that it seemed to fill the cerebral cavity unlike other ‘reptiles’ and that more details were thus visible, including the median division of the cerebrum. This close appression of the brain to the endocranial cavity characterizes many extinct members of the archosaur total-group, including extant birds and their stem-lineage (e.g., Franzosa & Rowe, 2005) and members of the archosaurian stem-lineage (e.g., Sobral et al., 2016), but appears to have been lost in extant crocodylians, which have a thick dural envelope (Hopson, 1979). More than a century after Seeley’s ( 1880) work, Wharton (2000) described a three-dimensional (3D) reconstruction of a teleosauroid endocast. Making an endocranial silicone rubber mould from an acid prepared specimen of Steneosaurus, Wharton noted the presence of an enlarged longitudinal dorsal venous sinus as compared to the modern gharial. However, the endocast mould was susceptible to the degree of acid preparation and thus incomplete. More recently, Brusatte et al. (2016) presented the braincase endocast of a second Steneosaurus specimen, confirming some of Seeley (1880) and Wharton’s (2000) observations and documenting some novel features such as the morphology of the endosseous labyrinth.

The endocranial characteristics of metriorhynchoids have also been investigated, with the majority of descriptions based on naturally preserved endocasts of metriorhynchids. Most attention has been placed on the presence of a hypertrophied, lobate salt-section gland, which sits within the enlarged prefrontal area and appears to drain into the antorbital fenestra (Fernández & Gasparini, 2000; Gandola et al., 2006; Fernández & Gasparini, 2008; Fernández & Herrera, 2009; Herrera, Fernández & Gasparini, 2013). More recently, neuroanatomical features from a natural endocast have been detailed (Herrera & Vennari, 2015), in addition to a virtual 3D reconstruction of the endocranial cavity in the metriorhynchid Cricosaurus araucanensis (Herrera, Fernández & Gasparini, 2013; Herrera, 2015). Similar to observations of teleosauroids, these studies found that the dural envelope surrounding the cerebral hemispheres was thin in metriorhynchoids and that the cerebrum filled the majority of the cranial cavity (e.g., Herrera, 2015; Herrera & Vennari, 2015). Also, in keeping with Wharton (2000) and Brusatte et al. (2016), there appears to be an enlarged dorsal venous sinus overlying the dorsal region of the hindbrain that appears to connect with the paratympanic sinus system (Fernández et al., 2011; Herrera, 2015). Further, the 3D reconstruction exposed an expanded antorbital sinus and a paired olfactory structure positioned between the salt-glands, features that appear, so far, to be unique to metriorhynchoids.

To further investigate endocranial anatomy in thalattosuchians, we examined a virtual 3D endocast of Pelagosaurus typus, a Toarcian (Early Jurassic) monotypic genus known from shallow marine sediments of Western Europe (Eudes-Deslongchamps, 1863; Eudes-Deslongchamps, 1877; Westphal, 1961; Westphal, 1962; Duffin, 1979; Pierce & Benton, 2006). The evolutionary affinity of P. typus has been a point of contention, with the species being diagnosed as both a teleosauroid (Eudes-Deslongchamps, 1863; Eudes-Deslongchamps, 1877; Westphal, 1961; Westphal, 1962; Duffin, 1979; Gasparini, Pol & Spalletti, 2006; Mueller-Töwe, 2005; Mueller-Töwe, 2006; Sereno & Larsson, 2009; Wilberg, 2015b) and a metriorhynchoid (Buffetaut, 1980; Buffetaut, 1982; Vignaud, 1995; Pol, Turner & Norell, 2009; Young et al., 2013; Adams, 2013; Parrilla-Bel et al., 2013; Wilberg, 2015a) or as the sister group to both clades (Benton & Clark, 1988; Clark, 1994; Buckley et al., 2000; Wu, Russell & Cumbaa, 2001; Tykoski et al., 2002; Pol, 2003). Pelagosaurus typus was a small-bodied thalattosuchian (∼1 m in length) considered to be an adept aquatic pursuit predator, with a long streamlined snout ideal for snapping at fast moving prey (e.g., fish) and large, anterolaterally placed orbits for increased visual acuity (Pierce & Benton, 2006). Its overall gross morphology has been extensively documented (e.g., Pierce & Benton, 2006), but little is known about its soft-tissue anatomy. Here we extend our understanding of P. typus by illustrating its endocranial anatomy and use this information to discuss potential functional paleoneurology, as well as endocranial evolution in thalattosuchians and crocodile-line archosaurs (pseudosuchians plus phytosaurs) more generally.

Material and Methods

We reconstructed the endocranial cavity of a three-dimensionally preserved skull of Pelagosaurus typus (specimen M1413) from the Charles Moore Collection housed in the Bath Royal Literary and Scientific Institute (BRLSI) (Figs. 1A and 1B; see Fig. S1 for interactive 3D reconstruction). The specimen comes from Strawberry Bank, north of Ilminster, Somerset England, is geologically from the lower part of the Upper Lias (Toarcian; Early Jurassic), and is preserved in a limestone nodule providing excellent 3D preservation with minimal taphonomic distortion. The external surface has been manually prepared, but the internal cavities are still infilled with limestone. As both the geological setting and the external skull morphology of this specimen have been described in detail by Pierce & Benton (2006), we focus solely on endocranial morphology here. For comparison, we also reconstructed the endocranial cavities of the extant crocodylian, Gavialis gangeticus (specimen R5792, University Museum of Zoology, Cambridge or UMZC) (Figs. 1C and 1D; see Fig. S2 for interactive 3D reconstruction). We chose Gavialis because out of all extant crocodylian species its skull morphology (longirostrine) and ecology (aquatic, piscivorus) are most analogous to that of Pelagosaurus. Furthermore, no detailed 3D endocranial reconstruction of a gharial is currently available in the literature (except for the vestibular system; Georgi & Sipla, 2008; Brusatte et al., 2016), so in addition to being a comparison for this study, it also provides valuable information about extant crocodylian internal skull morphology.

Figure 1 Three-dimensional reconstruction of the skull and underlying endocranial morphology.

(A–B) Pelagosaurus typus (BRLSI M1413); and (C–D) Gavialis gangeticus (UMZC R 5792). The star indicates the position of the antorbital fenestra in Pelagosaurus.

To get at the internal morphology of the specimens, the Pelagosaurus specimen M1413 was µCT scanned at the University of Texas at Austin’s High-Resolution X-ray CT Facility in two parts: the rostrum, which has a natural break about 2/3 from the tip of the snout and the posterior part of the skull which contains the braincase. When the two pieces are fit together the total skull length (from tip of snout to end of parietal table) is 270 mm. Resolution of each image is 1,024 ×1,024 pixels with an isotropic voxel size of 0.156 mm. The Gavialis specimen R5792 was a large adult male (skull length = 645 mm) and scanned on a medical CT scanner at The Royal Veterinary College, London. Resolution of each image is 512 ×512 pixels, the pixel width and height is 0.791 mm, and the voxel depth is 0.625 mm. Finally, 3D reconstructions of the brain and sinus cavities were done in Materialise Mimics® Research edition version 19 and rendered in Autodesk® 3ds Max 2015. Interactive 3D PDFs of the anatomical reconstructions are provided as supplementary figures to this paper. CT data for Pelagosaurus typus is stored on DigiMorph (http://digimorph.org/specimens/Pelagosaurus_typus/whole/), while that for Gavialis gangeticus has been reposited in the UMZC.

Figure 2 Morphometric measurements shown on illustrations of the endocast and endosseous labyrinth of Pelagosaurus.

(A–C) endocast shown in: (A) lateral view; (B) ventral view; (C) dorsal view. (D–E) left labyrinth shown in: (D) lateral view; (E) dorsal view. Abbreviations: AA, anterior semicircular canal area; CF, cephalic flexure angle; CW, maximum width of cerebrum; ECL, endosseous cochlea length; EL, total length of endocast; LA, lateral semicircular canal area; LH, maximum height of labyrinth; LW, maximum width of labyrinths; OL; length of olfactory tract (+blubs); PA, posterior semicircular canal area; PF, pontine flexure angle; PH; pituitary height; PL; pituitary length; PW, pituitary width. Anterior is towards the left, except in (E) where it is pointing down. See more detailed figures for anatomical identifications and size.

Table 1 Raw morphometric data.

Measurements were collected on pseudosuchian (phytosaurs and crocodile-line archosaurs) endocast and labyrinth reconstructions currently available in the literature (see table for sources), in addition to Pelagosaurus typus and Gavialis gangeticus (this study). All data are reported in mm or degrees and were measured in ImageJ (http://imagej.nih.gov/ij). Question marks denote missing data. The measurement protocol can be found in Fig. 1.

(Rounded to nearest mm)	Ebrachosaurus (phytosaur)	Parasuchus (phytosaur)	Pseudopalatus (phytosaur)	Desmatosuchus (aetosaur)	Pelagosaurus (thalattosuchian)	Steneosaurus (teleosauroid)	
Skull width at cerebrum (b/w postorbitals) (SW)	101	78	150	?	52	?	
Cephalic flexure angle (CF)	147	137	133	132	160	175	
Pontine flexure angle (PF)	146	136	141	131	160	170	
Endocast length (EL)	100	95	138	117	57	?	
Olfactory tract length (+bulbs) (OL)	53	47	71	30	21	?	
Cerebrum width (CW)	18	19	20	34	15	28	
Pituitary width (PW)	?	?	?	?	6	14	
Pituitary height (PH)	12	11	?	11	7	12	
Pituitary length (PL)	7	8	?	12	10	17	
Labyrinth height (LH)	14	14	?	?	14	26	
Labyrinth width (LW)	18	18	?	?	11	26	
Endosseous Cochlear duct length (ECL)	5	6	?	?	8	13	
Anterior semicircular canal area (AA)	23	16	?	?	9	38	
Posterior semicircular canal area (PA)	18	9	?	?	6	19	
Lateral semicircular canal area (LA)	8	11	?	?	4	14	
Source	Lautenschlager & Butler (2016)	Lautenschlager & Butler (2016)	Holloway, Claeson & O’Keefe (2013)	Hopson (1979)	This study	Brusatte et al. (2016)	
(Rounded to nearest mm)	Simosuchus (notosuchian)	Sebecus (sebecid)	Pholidosaurus (pholidosaur)	Goniopholis (goniopholid)	Gavialis (crocodylian)	Alligator (crocodylian)	Crocodylus (crocodylian)	
Skull width at cerebrum (b/w postorbitals) (SW)	58	147	?	?	168	73	?	
Cephalic flexure angle (CF)	142	150	143	140	150	135	145	
Pontine flexure angle (PF)	165	160	150	161	154	145	153	
Endocast length (EL)	79	120	138	117	146	98	103	
Olfactory tract length (+blubs) (OL)	25	46	51	42	55	48	46	
Cerebrum width (CW)	25	30	28	31	32	21	29	
Pituitary width (PW)	5	?	12	15	6	5	5	
Pituitary height (PH)	9	9	9	?	9	8	8	
Pituitary length (PL)	10	8	20	?	11	10	11	
Labyrinth height (LH)	?	?	?	?	21	18	13	
Labyrinth width (LW)	?	?	?	?	21	14	14	
Endosseous Cochlear duct length (CL)	?	?	?	?	9	8	6	
Anterior semicircular canal area (AA)	?	?	?	?	36	35	18	
Posterior semicircular canal area (PA)	?	?	?	?	15	12	5	
Lateral semicircular canal area (LA)	?	?	?	?	22	13	8	
Source	Kley et al. (2010)	Colbert (1946) and Hopson (1979)	Edinger (1938) and Hopson (1979)	Edinger (1938)	This study	Witmer & Ridgely (2008)	Witmer et al. (2008)	

Table 2 Comparison of endocast and labyrinth proportions in pseudosuchians (phytosaurs and crocodile-line archosaurs).

Ratios highlight proportions of the olfactory tract, cerebrum, pituitary, and endosseous labyrinth, and are calculated from Table 1. Question marks denote missing data. For anatomical abbreviations, see Fig. 1 and Table 1.

	Ebrachosaurus (phytosaur)	Parasuchus (phytosaur)	Pseudopalatus (phytosaur)	Desmatosuchus (aetosaur)	Pelagosaurus (thalattosuchian)	Steneosaurus (teleosauroid)	
CW:SW	0.18	0.24	0.13	?	0.29	?	
CW:EL	0.18	0.20	0.14	0.29	0.26	?	
OL:EL	0.53	0.50	0.51	0.26	0.37	?	
PW:PH	?	?	?	?	0.86	1.16	
PW:PL	?	?	?	?	0.60	0.82	
PL:(EL-OL)	0.15	0.17	?	0.18	0.28	0.23	
LW: LH	1.29	1.29	?	?	0.79	1.00	
ECL:LH	0.39	0.43	?	?	0.55	0.50	
AA:PA	1.25	1.80	?	?	1.50	2.00	
AA:LA	2.84	1.44	?	?	2.25	2.71	
PA:LA	2.27	0.80	?	?	1.50	1.36	
	Simosuchus (notosuchian)	Sebecus (sebecid)	Pholidosaurus (pholidosaur)	Goniopholis (goniopholid)	Gavialis (crocodylian)	Alligator (crocodylian)	Crocodylus (crocodylian)	
CW:SW	0.43	0.20	?	?	0.19	0.13	?	
CW:EL	0.32	0.25	0.20	0.26	0.22	0.22	0.28	
OL:EL	0.32	0.38	0.37	0.36	0.38	0.49	0.45	
PW:PH	0.56	?	1.33	?	0.67	0.63	0.63	
PW:PL	0.50	?	0.6	?	0.55	0.50	0.45	
PL:(EL-OL)	0.19		0.23	?	0.12	0.20	0.19	
LW: LH	?	?	?	?	1.00	0.78	1.05	
ECL:LH	?	?	?	?	0.43	0.44	0.45	
AA:PA	?	?	?	?	2.39	2.88	3.68	
AA:LA	?	?	?	?	1.63	2.82	2.16	
PA:LA	?	?	?	?	0.68	0.98	0.59	

In addition to details of gross morphology of the endocranium, we also captured morphometric data (Fig. 2; Table 1) on the endocasts of Pelagosaurus, Gavialis, and a selection of documented pseudosuchians (including phytosaurs) from the literature, including: the phytosaurs Ebrachosaurus neukami (Lautenschlager & Butler, 2016), Parasuchus (=Paleorhinus) angustifrons (Lautenschlager & Butler, 2016), and Machaeroprosopus (=Pseudopalatus) mccauleyi (Holloway, Claeson & O’Keefe, 2013); the aetosaur Desmatosuchus spurensis (Hopson, 1979); the teleosauroid Steneosaurus cf. gracilirostris (Brusatte et al., 2016); the notosuchian Simosuchus clarki (Kley et al., 2010); the sebecid Sebecus icaeorhinus (Colbert, 1946; Hopson, 1979); the pholidosaur Pholidosaurus meyeri (Edinger, 1938; Hopson, 1979); the goniopholid Goniopholis pugnax (Edinger, 1938); and the extant crocodylians Alligator mississippiensis (Witmer & Ridgely, 2008) and Crocodylus johnstoni (Witmer et al., 2008). Phytosaurs are included here as they either represent the earliest diverging pseudosuchians (Brusatte et al., 2010; Ezcurra, 2016) or are the sister-group to the Archosauria (Nesbitt, 2011); in either case, phytosaurs provide information on the plesiomorphic condition for Pseudosuchia. Further, we refrained from collecting morphometric data on the teleosauroid thalattosuchians Teleosaurus eucephalus (Seeley, 1880) and Steneosaurus pictaviensis (Wharton, 2000) and the metriorhynchoid Cricosaurus (=Geosaurus) araucanensis (Herrera, Fernández & Gasparini, 2013) as the endocasts are incomplete and not preserved with enough detail. These species are, however, used in a comparative context. For inter-taxon comparison, the raw morphometric data (Fig. 2; Table 1) were converted into ratios that highlight proportions of the olfactory tract, cerebrum, pituitary fossa, and endosseous labyrinth (see Table 2 for details).

Description

Nasal cavity and associated structures

A number of major features are shared by Pelagosaurus typus and Gavialis gangeticus. Both have an elongate nasal cavity (Figs. 3 and 4; dark yellow) that extends posteriorly from the external naris (=primary choana) at the tip of the snout to the retracted internal naris (=secondary choana), ventral to the basicranium. Thus, a secondary palate is present in both taxa, and the nasal passage spans almost the entire length of the skull. The external nares form a single, midline opening in both taxa, as seen in various crocodyliforms, including crocodylids, Isisfordia (Salisbury et al., 2006), pholidosaurids/dyrosaurids (e.g., Sereno et al., 2001; Jouve, 2005) and Goniopholis (Holland, 1905), as well as in other thalattosuchians (e.g., Fraas, 1901; Andrews, 1913; Gasparini, Pol & Spalletti, 2006). However, given the likely basal phylogenetic position of Thalattosuchia among crocodyliforms (e.g., Wilberg, 2015a), and the presence of paired external nares and unretracted choanae in many phylogenetically intermediate crocodyliforms (e.g., Ortega et al., 2000; Clark & Sues, 2002; Carvalho, Ribeiro & Avilla, 2004; Kley et al., 2010), these features shared by Gavialis and Pelagosaurus represent independent evolutionary acquisitions, probably reflecting adaptation to aquatic life.

Figure 3 Reconstruction of the endocranial anatomy of Pelagosaurus typus.

(A) anterior view; (B) posterior view; (C) dorsal view; (D) lateral view; (E) ventral view. Abbreviations: Bod, basioccipital diverticulum; BSd, basisphenoid diverticulum; EC, external choana; END, endocast; EL, endosseous labyrinth; IC, internal choana; MPS, median pharyngeal sinus; NC, nasal cavity; NPD, nasopharyngeal duct; OR, olfactory region of the nasal cavity; PNS, paranasal sinus; PTS, pharyngotympanic sinus; VC, neurovascular canal; VS, venous sinus. Scale bars equal 1 cm.

Figure 4 Reconstruction of the endocranial anatomy of Gavialis gangeticus.

(A) anterior view; (B) posterior view; (C) dorsal view; (D) lateral view; (E) ventral view. Abbreviations: BL, bulla; EAM, external auditory meatus; EC, external choana; END, endocast; IC, internal choana; ITS, intertympanic sinus; MPS, median pharyngeal sinus; NC, nasal cavity; NLD, naso-lacrimal duct; NPD, nasopharyngeal duct; OR, olfactory region of the nasal cavity; PAS, parietal sinus; PNS, paranasal sinus; PTS, pharyngotympanic sinus; PTT, pharyngotympanic tube; QS, quadrate sinus; VC, neurovascular canal. Scale bars equal 1 cm.

The nasal cavity of Gavialis exhibits three derived features that are absent in Pelagosaurus. (1) Anteriorly, the nasal passageway of Gavialis is inflected abruptly dorsally (Fig. 4D; dark yellow), forming an expanded cylindrical recess that communicates externally via a dorsally-placed external naris; dorsal inflection of the external naris also characterizes other eusuchians such as Alligator mississippiensis (Witmer & Ridgely, 2008). In Pelagosaurus, the external naris faces anterodorsally (Fig. 3D; dark yellow), a morphology shared with other thalattosuchians (e.g., Andrews, 1913) and longirostrine crocodylomorphs, such as pholidosaurids/dyrosaurids (Sereno et al., 2001; Jouve, 2005). (2) The internal naris of Gavialis (Figs. 4D and 4E; dark yellow) is enclosed by the pterygoids, is located ventral to the labyrinth at the posterior end of the temporal fossa (as seen in other neosuchians; Salisbury et al., 2006; Turner & Buckley, 2008), and the cross-sectional area of the nasal passageway here is expanded (perhaps incorporating part of the pterygoid sinus). This contrasts with the relatively unexpanded posterior region of the nasal passageway of Pelagosaurus, and the more anterior location of the internal naris, enclosed primarily by the palatines, and situated ventral to the cerebrum at the anterior end of the subtemporal fossa (Figs. 3D and 3E; dark yellow). The relatively anterior placement of the internal naris of Pelagosaurus, and its primary enclosure by the palatines, is widespread among thalattosuchians (e.g., Andrews, 1913) and other non-eusuchian crocodylomorphs (Salisbury et al., 2006; Turner & Buckley, 2008). (3) The pterygoid of Gavialis is pneumatised by a diverticulum of the nasal epithelium, forming an expanded bulla that is absent in Pelagosaurus (Fig. 4; light green). This bulla is only present in large individuals of Gavialis, and is proposed to have a function in vocalization (Martin & Bellairs, 1977).

Immediately anterior to the endocast, the nasal passageway is expanded to form the olfactory region (dorsal region of the cavum nasi proprium; Parsons, 1970) in both specimens (Figs. 3 and 4; dark green). This is dorsal to, and distinct from, the posteroventral continuation of the nasal passageway, which is paired in this region in both taxa forming the nasopharyngeal duct (Figs. 3E and 4E; dark yellow). In Gavialis, the central portion of the olfactory region bears a weak midline groove, and is relatively smaller than in Pelagosaurus, being both anteroposteriorly shorter and mediolaterally narrower (Figs. 4A and 4C; dark green). Conversely, the central portion of the olfactory region of Pelagosaurus bears a deep midline dorsal groove, creating bilaterally symmetrical, bulbous expansions (Figs. 3A and 3C; dark green). A similar set of olfactory recesses to that seen in Pelagosaurus is present in the metriorhynchoid Cricosaurus (Fernández & Herrera, 2009; Herrera, Fernández & Gasparini, 2013). In Cricosaurus, the large, bulbous dorsal region is occupied by an apparent soft tissue structure comprised of ‘lobules’. This structure has been reported in multiple natural endocasts of metriorhynchoids (e.g., Fernández & Gasparini, 2000; Fernández & Gasparini, 2008; Herrera, 2015), and has been interpreted as housing enlarged salt glands, which may exit through the antorbital fenestra. Given their identical topology, we propose the enlarged bulbous recesses seen in Pelagosaurus to be osteological correlates of the same structures, suggesting the presence of antorbital salt glands in one of the earliest diverging thalattosuchians. This interpretation does not preclude the presence of other structures, including olfactory and pneumatic epithelia, such as portions of the paranasal sinus, within the olfactory recess. Therefore, we continue using the term ‘olfactory region’, consistent with the homology of this recess in taxa that lack large salt glands (Parsons, 1970). In other words, we suggest that the preorbital recess for a hypothesized salt gland in thalattosuchians is an expansion of the olfactory region of other reptiles.

In many archosaurs, the paranasal sinus system perforates the lateral surface of the skull, forming a large antorbital fenestra between the maxilla, nasal and lacrimal (Witmer, 1997). Although the antorbital fenestra is closed in extant crocodylians such as Gavialis, the internal paranasal sinus is still well developed (Witmer, 1997). Unlike in crocodylians, the antorbital fenestra is small, but apparently present in Pelagosaurus (Witmer, 1997; Pierce & Benton, 2006). Witmer (1997) described Pelagosaurus as having a small, slit-like antorbital fenestra between the maxilla and lacrimal, with little or no surrounding external fossa, and this is evident in the specimen described here (Fig. 1A; and Pierce & Benton, 2006). This condition is similar to early teleosauroids, such as Teleosaurus, which also have an antorbital fenestra that is small, but nevertheless present (Jouve, 2009; the fenestra is closed in Machimosaurus, Martin & Vincent, 2013). A different morphology is present in metriorhynchoid thalattosuchians (reviewed by Leardi, Pol & Fernández, 2012: Fig. 1), in which a larger, circular fenestra is present in this region, from which a broad groove extends anteriorly along the lateral surface of the snout, and which has been interpreted as being homologous with the antorbital fenestra of other archosaurs by most authors (e.g., Andrews, 1913; Witmer, 1997; Gasparini, Pol & Spalletti, 2006; Young & De Andrade, 2009). Because of ambiguity over the homology of the ‘antorbital fenestra’ in thalattosuchians generally, and in metriorhynchoids specifically, Fernández & Herrera (2009) and Leardi, Pol & Fernández (2012) advocated use of the term ‘preorbital fenestra’ to describe this feature. In Pelagosaurus, this fenestra enters internally onto the dorsolateral surface of the olfactory recess anteriorly (the position of the antorbital fenestra is indicated by a star in Figs. 1A–1B).

Although the preorbital fenestra of thalattosuchians has been identified as a reduced external antorbital fenestra by many authors (e.g., Witmer, 1997), Fernández & Herrera (2009) proposed an alternative hypothesis. Based on the observed association of this external opening with the internal recess for a salt gland (labeled here as ‘olfactory region’; Fig. 4), Fernández & Herrera (2009) suggested that the external opening of the antorbital fenestra was closed in metriorhynchids, and their external opening instead represented a neomorphic exit for the salt gland. This hypothesis was supported by dynamic homology analyses by Leardi, Pol & Fernández (2012) for metriorynchids, but not for other thalattosuchians. The analysis of Leardi, Pol & Fernández (2012) requires reassessment in light of the topological similarity of structures associated with the preorbital fenestra (or antorbital fenestra) in Pelagosaurus to those of metriorhynchids. This raises the possibility that our ‘antorbital fenestra’ instead represents an external opening of the salt gland. For now, we denote the external opening as the antorbital fenestra here. In reconciliation of these hypotheses, we see no reason why the internal recess (‘olfactory’ region) might not have housed an enlarged salt gland, alongside olfactory epithelia and portions of the internal paranasal sinus, or why the external opening might not have provided an exit for the salt gland as well as being homologous with the antorbital fenestra of other archosaurs.

In Pelagosaurus and Gavialis, there is a subconical, subsidiary outpocketing from the main portion of the olfactory region ventrolateral to the nasal passageway (Figs. 3D, 3E, 4D and 4E; dark green). We identify this outpocketing as part of the paranasal sinus system, which likely represents the antorbital sinus (Fernández & Herrera, 2009). A similar feature has been identified as the antorbital sinus in the metriorhynchoid Cricosaurus (Fernández & Herrera, 2009; Herrera, Fernández & Gasparini, 2013) and phytosaurs also have an enlarged antorbital sinus in this region (Lautenschlager & Butler, 2016). In both Pelagosaurus and Gavialis, the hypothesized antorbital sinus (i.e., the hypothesized internal recess for the major portion of the paranasal sinus) is confluent anteriorly with elongate internal canals that extend longitudinally along the length of the snout (Figs. 3 and 4; dark pink). Despite the confluence of these recesses, we identify the tapering anterior portions as neurovascular canals rather than being the anterior portion of the antorbital sinuses. In particular, we identify it as the dorsal alveolar canal for the maxillary branch of the trigeminal nerve and maxillary vein and artery. As added evidence of these, we have observed smaller canals that branch off and enter the bases of the maxillary and premaxillary alveoli, indicating a neurovascular function. In addition to the features described above, Gavialis also has a sinus above the olfactory region, here identified as the naso-lacrimal duct (Figs. 4A, 4C and 4D; dark orange) as it occupies a similar position to the naso-lacrimal duct in Alligator (Witmer, 1997); this duct is not distinguishable in Pelagosaurus. In overall morphology, the paranasal sinus system as seen in Pelagosaurus and Gavialis is simplified as compared to the condition in brevirostrine, broad-snouted crocodiles, such as Alligator, in which the sinus forms broad, mediolateral pockets along the snout, and in many non-crocodylomorph archosaurs with open antorbital fenestrae, in which the sinus also forms a large, broad recess (Witmer, 1997; Witmer & Ridgely, 2008).

Endocranial cast

In overall appearance, the endocast of Pelagosaurus typus is similar to Gavialis gangeticus and other crocodile-line archosaurs (Edinger, 1938; Colbert, 1946; Hopson, 1979; Witmer & Ridgely, 2008; Witmer et al., 2008; Kley et al., 2010; Herrera, Fernández & Gasparini, 2013), including phytosaurs (Holloway, Claeson & O’Keefe, 2013; Lautenschlager & Butler, 2016), being approximately ‘cylindrical’ in form (Fig. 5). In Pelagosaurus, the brain endocast (vol = 4,045 mm3) appears proportionally larger than that of Gavialis, and is relatively more straight in outline. The angle of the cephalic (forebrain-midbrain) and pontine (midbrain-hindbrain) flexure in Pelagosaurus is much greater (i.e., less acute) than most other pseudosuchians, indicating that the brain is relatively straight (Table 1). A straight brain is shared with the metriorhynchoids Dakosaurus cf. andiniensis (Herrera & Vennari, 2015), Cricosaurus araucanensis (Herrera, 2015) and Metriorhynchus cf. westermanni (Fernández et al., 2011), and the teleosauroid Steneosaurus cf. gracilirostris (Table 1); the sectioned braincase of the teleosauroid Teleosaurus eucephalus also appears more straight in outline (Seeley, 1880). This may suggest that a straight brain endocast is a derived feature of thalattosuchians. The degree of flexion in avian brain endocasts is determined by the position and morphology of the orbit (Kawabe, Ando & Endo, 2014), and it is possible that the more lateral position of the orbit in thalattosuchians (e.g., Andrews, 1913; Gasparini, Pol & Spalletti, 2006; Martin & Vincent, 2013) may explain the reduction of brain endocast flexion in these taxa.

Figure 5 Endocast morphology.

(A–C) endocast of Pelgaosaurus typus and (D–F) endocast of Gavialis gangeticus. (A–D), dorsal view; (B–E), lateral view; (C–F), ventral view. Abbreviations: BA, basilar artery; CBL, cerebellum; CER, cerebrum; DLS, dorsal branch of longitudinal sinus; ICa, internal carotid; MO, medulla oblongata; OA, orbital artery; OT, olfactory tract and bulb; PIT, pituitary; II, optic nerve region; III, oculomotor nerve region; V, trigeminal nerve region; VII, facial nerve region; VIII, vestibulocochlear nerve; IX–XI, glossopharyngeal, vagus, and accessory nerve region; XII, hypoglossal nerve region. For visualization, the endocast of Gavialis has been scaled to the same anteroposterior length as Pelagosaurus. Scale bars equal 1 cm.

Starting anteriorly in the forebrain endocast, the olfactory tract and bulbs in Pelagosaurus are straight and take the form of a pair of tapering, anteriorly directed finger-like extensions that merge anteriorly with the olfactory region of the narial cavity (Figs. 5A–5C). This is in contrast to Gavialis, which has an anteroventrally directed olfactory tract and does not have an osteological division between the olfactory bulbs (Figs. 5D–5F), a morphology similar to other extant crocodylians (Witmer & Ridgely, 2008; Witmer et al., 2008), phytosaurs (Holloway, Claeson & O’Keefe, 2013; Lautenschlager & Butler, 2016), and various crocodylomorphs (Edinger, 1938; Hopson, 1979); although, the fossa for the olfactory bulb is partially divided in Sebecus (Colbert, 1946; Hopson, 1979) and is proportionally larger in the notosuchian Simosuchus clarki (Kley et al., 2010). The aetosaur Desmatosuchus has a proportionally even larger pair of olfactory bulbs that likely represents an independently derived feature (Hopson, 1979). The olfactory bulb in the metriorhynchoid Cricosaurus is also undivided, but attach to a paired olfactory region in the snout (Herrera, Fernández & Gasparini, 2013). Such a paired structure may also be present in the olfactory region of Pelagosaurus, as the posterodorsal portion of this region sends off a pair of rami that connect with the olfactory bulbs posteriorly (Figs. 3C and 3D; green posterior extensions from olfactory region). In terms of proportions, the olfactory tract (plus blubs) in Pelagosaurus is similar in size to other crocodile-line archosaurs, with the exception of phytosaurs and Cricosaurus, which have elongated olfactory tracts that form approximately half the length of the endocast (Table 2).

Visually, the cerebrum of Pelagosaurus is laterally expanded and bulbous compared to that of Gavialis (Fig. 5). In fact, compared to other crocodile-line archosaurs, the cerebrum of Pelagosaurus is proportionally larger compared to skull width than in any taxon other than the notosuchian Simosuchus (Table 2). The outline of the cerebrum in Pelagosaurus in dorsal view is symmetrical along its length (Fig. 5A), whereas the cerebrum of Gavialis, phytosaurs (Holloway, Claeson & O’Keefe, 2013; Lautenschlager & Butler, 2016) and other crocodylomorphs (Edinger, 1938; Colbert, 1946; Hopson, 1979; Kley et al., 2010) is most strongly expanded posteriorly (Fig. 5D). The dorsal surface of the cerebral cast bears a shallow, midline groove in Pelagosaurus, indicating the division between hemispheres by the cerebral longitudinal fissure, resulting in a heart-shaped cross-section (Fig. 5A). This groove is absent in Gavialis (Fig. 5D) and most other reptiles (Witmer & Ridgely, 2008; Witmer et al., 2008; Kley et al., 2010; Lautenschlager & Butler, 2016), including the metriorhynchoid Cricosaurus (Herrera, 2015) and the teleosauroid Steneosaurus pictaviensis (Wharton, 2000). However, Seeley (1880) alluded to the division of the cerebrum in Teleosaurus. The absence of this groove occurs because the anterior portion of the dorsal longitudinal sinus, which extends along the central portion of the brain, is covered by a thick dural envelope (Hopson, 1979). The appearance of the division between the cerebral hemispheres in the brain endocast of Pelagosaurus (and potentially Teleosaurus) is therefore unusual among pseudosuchians studied so far, and may suggest that the dural envelope surrounding the cerebral hemispheres was relatively thin in this taxon. Such a morphology is common within some dinosaur clades (e.g., hadrosauroids) in which the cast of the cerebral hemispheres also bears vascular impressions (Evans, 2005).

One of the most striking features in the brain endocast of Pelagosaurus, as compared to Gavialis and other crocodile-line archosaurs (Hopson, 1979; Kley et al., 2010; Lautenschlager & Butler, 2016), is the greatly enlarged pituitary, which emerges posteroventrally from the cerebrum just anterior to the optic lobe region (Fig. 5). In Pelagosaurus, the pituitary is anteroposteriorly long and is also proportionately wide as compared to its overall depth (Table 2). The pituitary is also characterized by two distinct anterodorsally projecting channels (see further below), and large posterolaterally projecting channels that housed the two branches of the internal carotid artery (Figs. 5B and 5D); the channels for the internal carotid artery curve dorsolaterally (and eventually posteroventrally) in Gavialis and other extant crocodylians (Hopson, 1979; Witmer et al., 2008; Dufeau & Witmer, 2015). The pituitary described by Seeley (1880) for the teleosauroid Teleosaurus also appears to be anteroposteriorly expanded, at least in the sagittal view available, and that of the teleosauroid Steneosaurus is similar to Pelagosaurus (Table 2; the pituitary of Pholidosaurus is also relatively large). Although Brusatte et al. (2016) described the pituitary fossa of Steneosaurus as being similar to that of extant crocodylians, our measurements indicate an anteroposteriorly enlarged pituitary fossa in Steneosaurus (Table 2). The endocast of Steneosaurus prepared by Wharton (2000) appears to have a pituitary of similar dimensions to that of Gavialis; however, the morphology of this area is not well preserved in the silicone endocast. Unfortunately, the pituitary fossa of the metriorhynchoid Cricosaurus was not reported by Herrera, Fernández & Gasparini (2013), and visualizations of this structure in “Metriorhynchus” cf. westermanni by Fernández et al. (2011) are not sufficiently clear to determine the morphology. Therefore, a large and anteroposteriorly expanded pituitary fossa may be a synapomorphy of Thalattosuchia, but further data on metriorhynchoids is necessary to fully support this conclusion.

As is typical of crocodile-line archosaurs, the optic lobes (of the midbrain) are not well delimited due to the thick dural envelope in this region (Hopson, 1979). However, the hindbrain, composed of the cerebellum and medulla oblongata (and embracing the vestibular system and cranial nerves V–XII), is distinguishable. In keeping with the forebrain, the cerebellum is expanded in Pelagosaurus compared to that of Gavialis (Figs. 5A and 5D); in fact, the cerebellum of Gavialis seems to be smaller than that of other extant crocodylians, including Alligator (Witmer & Ridgely, 2008), Crocodylus johnstoni (Witmer et al., 2008), and Caiman crocodilus (Hopson, 1979). In contrast to Gavialis, the cerebellum region in Pelagosaurus is characterized by two large, dorsally projecting rami, which are identified as branches of the dorsal longitudinal venous sinus (Figs. 5A and 5B). These branches, which presumably housed the caudal middle cerebral (head) vein (Witmer & Ridgely, 2008; Porter, 2015; Porter, Sedlmayr & Witmer, 2016), connect the endocranial cavity with the paratympanic system via a venous sinus (Fig. 3, light blue). The sinus that surrounds the caudal middle cerebral vein does not connect to the paratympanic sinus system in other crocodile-line archosaurs, including phytosaurs, but was recently described in the braincase of the metriorhynchoids “Metriorhynchus” cf. westermanni (Fernández et al., 2011), Dakosaurus cf. andiniensis and Cricosaurus araucanensis (Herrera, 2015), and similar dorsally directed branches extending from the cerebellum region have been described for the teleosauroid Steneosaurus (Wharton, 2000; Brusatte et al., 2016), suggesting this could represent a thalattosuchian synapomorphy. Furthermore, the medulla oblongata is foreshortened posterior to the vestibular depression in Pelagosaurus as compared to Gavialis (Fig. 5) and other pseudosuchians (e.g., Hopson, 1979), although it is relatively broad in cross-section. There is also a ventral swelling on the medulla in Pelagosaurus, just at the posterior limit of the vestibular depression, which is here identified as part of the basal artery (Figs. 5B and 5C). Such a swelling was also described as the basal artery in the endocast of the extant Caiman (Hopson, 1979).

Due to the preservation of Pelagosaurus and low resolution of the scan in Gavialis, delicate features like the cranial nerves were not easily delineated during segmentation. Most of the cranial nerves in the two taxa appear to be originating from similar areas on the brain endocast (Fig. 5). However, there are two features of note. Firstly, large, paired channels emerge anterodorsally from the pituitary of Pelagosaurus (Figs. 5B and 5C; green). This morphology is absent from Gavialis and the brain endocasts of most archosaurs, both crocodile-line and bird-line (Hopson, 1979; Witmer et al., 2008). But similar large channels have been described extending anteriorly from the pituitary in the teleosauroids Teleosaurus (Seeley, 1880) and Steneosaurus (Wharton, 2000; Brusatte et al., 2016). These channels may therefore represent a thalattosuchian synapomorphy. Unfortunately, currently available data precludes us from assessing the character state in metriorhynchoids; however, such a feature may be present in Cricosaurus araucanensis (A Paulina Carabajal, pers. comm., 2017). Seeley (1880) identified these channels as housing the optic nerve (CN II), although this is likely an error, as the optic nerve is visible anteriorly and was mistakenly labeled as the olfactory nerve (CN I) in his figure (Seeley, 1880: pl. 24). Wharton (2000) labeled this feature as the oculomotor nerve (CN III) in her silicon mould, a more-likely identification as CN III is closely associated with the anterior region of the pituitary fossa (Hopson, 1979; Witmer et al., 2008). Most recently, Brusatte et al. (2016) suggested instead that these channels housed the orbital arteries (a branch of the cerebral carotid arteries). The second feature of note is the trigeminal nerve fossa (CN V). Compared to Gavialis and other extant crocodylians (Hopson, 1979; Witmer & Ridgely, 2008; Witmer et al., 2008) and pseudosuchians (Edinger, 1938; Hopson, 1979; Kley et al., 2010; Lautenschlager & Butler, 2016; Brusatte et al., 2016), CN V fossa is relatively small in Pelagosaurus (Figs. 5A–5C), with limited lateral projection and a small cross-section, which may indicate that the trigeminal ganglion is situated outside the braincase wall in this taxon.

Endosseous labyrinth

Several features distinguish the endosseous labyrinth of Pelagosaurus typus from that of Gavialis gangeticus (Fig. 6; Table 2). The aspect ratio (anteroposterior width:dorsoventral height) of the vestibular region is high in Gavialis (Fig. 6 and Table 2), and the anterior and posterior semicircular canals arc dorsally from the common crus in a smooth curve, such that any given segment of either canal is curved along its length, and resulting overall in an ‘m’-shape to the anterior and posterior canals when seen in lateral view (Fig. 6F). This is similar to the morphology seen in other crocodylians (e.g., Georgi & Sipla, 2008; Witmer et al., 2008; Dufeau & Witmer, 2015; Brusatte et al., 2016), crocodylomorphs (Kley et al., 2010), and phytosaurs (Lautenschlager & Butler, 2016) (Fig. 7). By contrast, in Pelagosaurus the anterior and posterior canals extend dorsally from the common crus, but are then strongly inflected near their apices, and then are approximately straight for most of their length (Fig. 6B). This confers a ‘pyramidal’ appearance to the labyrinth of Pelagosaurus, which is also present in the teleosauroid Steneosaurus (Brusatte et al., 2016), and may ultimately be found to be a synapomorphy of Thalattosuchia as it is absent in Euparkeria, phytosaurs, Simosuchus, and many extant crocodylians (Fig. 7). Nevertheless, the labyrinth varies among extant crocodylians (Brusatte et al., 2016) and some taxa have a pyramidal labyrinth morphology similar to those of Pelagosaurus and Steneosaurus, especially Crocodylus johnstoni (Fig. 7; redrawn from Brusatte et al., 2016), indicating homoplasy in this trait.

Figure 6 Endosseous labyrinth.

(A–D) left inner ear of Pelagosaurus typus and (E–H) left inner ear of Gavialis gangeticus. (A–E) anterior view; (B–F) lateral view; (C–G) posterior view; (D–H) dorsal view. Abbreviations: ASC, anterior semicircular canal; CC, common crus; CD, cochlear duct; LSC, lateral semicircular canal; PSC, posterior semicircular canal. For visualization, the labyrinth of Gavialis has been scaled to the same anteroposterior width as Pelagosaurus. Scale bars equal 1 cm.

Figure 7 Evolution of the labyrinth in pseudosuchians and proximate stem-group archosaurs.

Labyrinths in lateral view are redrawn from Sobral et al. (2016: Euparkeria, left labyrinth), Lautenschlager & Butler (2016: the phytosaur Parasuchus, left labyrinth), Brusatte et al. (2016: Steneosaurus and Crocodylus johnstoni, left labyrinths), Kley et al. (2010: the notosuchian Simosuchus, reversed right labyrinth). Abbreviations: ASC, anterior semicircular canal; CD, cochlear duct; LSC, lateral semicircular canal; PSC, posterior semicircular canal.

The path of the anterior semicircular canal is substantially longer than that of the posterior semicircular canal in Gavialis (Figs. 6F and 6H), and extends far anteriorly. This morphology is also widespread among extant and extinct crocodylians (Georgi & Sipla, 2008; Witmer & Ridgely, 2008; Witmer et al., 2008; Bona, Degrange & Fernández, 2013; Dufeau & Witmer, 2015; Brusatte et al., 2016), but differs from the more equal proportions of the anterior and posterior canals seen in Pelagosaurus (Figs. 6B and Fig. 6D; Table 2), Steneosaurus (Brusatte et al., 2016), the notosuchian Simosuchus (Kley et al., 2010), phytosaurs (Lautenschlager & Butler, 2016), and the crownward stem-group archosaur Euparkeria (Sobral et al., 2016) (Fig. 7; Table 2). Therefore, the larger size of the anterior canal in extant crocodylians is probably a derived feature of eusuchians or crocodylians, and future work will determine its functional significance and when it evolved on the crocodylian stem lineage. The size of the semicircular canals is considered proportional to their sensitivity (Sipla & Spoor, 2008) and the presence of a larger anterior canal in mammals (Spoor et al., 2007), including most cetaceans (Ekdale & Racicot, 2015), has been suggested to increase the sensitivity to pitch motions of the head (reviewed by Ekdale, 2015), but this seems unlikely to be true in crocodylians which have laterally undulating locomotion.

The length of the endosseus cochlear duct (=lagenar recess) sensu Walsh et al. (2009) is proportionally much longer in Pelagosaurus than in Gavialis or most other crocodile-line archosaurs (Fig. 6; Table 2), or even those of early dinosaurs (e.g., cochlear duct length:labyrinth length = ∼0.5 in both Herrerasaurus and Massospondylus; Sereno et al., 2007; Knoll et al., 2012). An elongated cochlear duct is also present in the teleosauroid Steneosaurus (Brusatte et al., 2016), and it is likely that this feature may be a synapomorphy of Thalattosuchia (Fig. 7). Elongation of the cochlear duct may indicate enhanced auditory capabilities given that dorsoventral length of the cochlea is correlated with acoustic capabilities in extant birds and reptiles (Wever, 1978; Gleich & Manley, 2000; Witmer et al., 2008; Walsh et al., 2009).

Paratympanic sinuses

The otic region in archosaurs is pneumatised by diverticula of the middle ear, forming paratympanic sinuses that function to enhance acoustic capabilities of the middle ear in extant crocodylians (Witmer & Ridgely, 2008; Dufeau & Witmer, 2015). The 3D morphology of the paratympanic sinuses has been described in Alligator mississippiensis (Witmer & Ridgely, 2008; Dufeau & Witmer, 2015) and Crocodylus johnstoni (Witmer et al., 2008); however, little is known about the system outside extant crocodylians (but see Bona, Degrange & Fernández, 2013). There are a number of differences between the paratympanic sinuses of Pelagosaurus typus (Fig. 3; dark purple) and extant crocodylians (see further below), but the morphology is very similar to that recently described for the metriorhynchoid “Metriorhynchus” cf. westermanni (Fernández et al., 2011). In Pelagosaurus, the paratympanic sinus is separated into two cavities or sinus channels (Figs. 3A and 3B). A dorsal (venous) sinus (cavity 1 of Fernández et al., 2011) is confluent with the dorsal longitudinal sinus on the endocast (Fig. 3; light blue) and extends ventrolaterally to join with the pharyngotympanic sinus, close to the external auditory meatus. This sinus presumably contained the caudal head vein (see above). Moving medially from the external auditory meatus in Pelagosaurus, there is a large pharyngotympanic sinus (cavity 2 of Fernández et al., 2011), that branches ventrally below the endocast (Figs. 3B, 3D–3E). One branch runs medially towards and anteroventrally underneath the vestibular system; at this point it wraps around the endocast ventrally and eventually forms the basisphenoid diverticulum (Dufeau, 2011; Dufeau & Witmer, 2015) (Figs. 3D and 3E). The second branch extends medially towards the vestibular system and then dives ventrally and wraps around the endocast posteriorly forming the basioccipital diverticulum (Dufeau, 2011; Dufeau & Witmer, 2015) (Figs. 3B, 3D and 3E). Ventrally, there is an anteroposterior directed median pharyngeal sinus (=median Eustachian tube; Colbert, 1946) that connects the anterior and posterior diverticula, respectively (Colbert, 1946; Dufeau, 2011; Dufeau & Witmer, 2015) (Figs. 3D and 3E). The pharyngotympanic tubes (= lateral Eustachian tubes; Colbert, 1946) are not visible here, but where identified by Dufeau (2011) as small posteroventral extensions of the pharyngotympanic sinus.

The morphology of the paratympanic sinus system in Pelagosaurus contrasts strongly with that in Gavialis (Fig. 4; dark purple) and other extant crocodylians (Witmer & Ridgely, 2008; Witmer et al., 2008; Dufeau & Witmer, 2015). Although Gavialis has a dorsal branch of the paratympanic sinus (Figs. 4A and 4B), which fills the posttemporal fenestra, it does not contact the endocast dorsally via the dorsal longitudinal sinus; however, it does run ventrolaterally to join the middle ear cavity (pharyngotympanic sinus). Moving medially from the external auditory meatus (which is relatively large as compared to Pelagosaurus), there is a large intertympanic sinus (composed of various diverticula; see Dufeau & Witmer, 2015) that connects the right and left middle ear cavities and runs dorsally around the endocast and contacts with the parietal sinus cranially (Fig. 4B). Branching ventrally from the pharyngotympanic sinus is the Eustachian system (Colbert, 1946; Dufeau & Witmer, 2015) (Figs. 4B and 4D). Unlike Pelagosaurus, the Eustachian system in Gavialis, and other extant crocodylians, extends ventrally causing the medial pharyngeal sinus and the basisphenoid/basioccipital diverticula to become vertically oriented (Colbert, 1946; Dufeau & Witmer, 2015). Further, the pharyngotympanic tubes (= lateral Eustachian tubes; Colbert, 1946) are ventrally elongated. Finally, in Gavilais the pharyngotympanic sinus sends a channel posteriorly down the medial aspect of the quadrate forming a quadrate sinus (Fig. 4B), which is not present in Pelagosaurus (also noted by Dufeau, 2011).

Discussion

The detailed anatomical observations presented here expand our knowledge of endocranial anatomy in early thalattosuchians, building on landmark studies of Late Jurassic members of the group (Fernández & Gasparini, 2000; Gandola et al., 2006; Fernández & Gasparini, 2008; Herrera, Fernández & Gasparini, 2013; Herrera & Vennari, 2015) and less complete data provided so far for some Early Jurassic teleosauroids (Seeley, 1880; Wharton, 2000; Brusatte et al., 2016). Based on our 3D endocranial reconstructions of the thalattosuchian Pelagosaurus typus and of the extant crocodylian Gavialis gangeticus, and making use of existing knowledge of the relationships of these taxa, we are able to clarify the phylogenetic distributions of key soft-tissue features of thalattosuchians, including features of the paranasal and paratympanic sinus systems, the neuroanatomy of the brain and vestibular organ, and potential physiological adaptations. Below, we propose functional interpretations relevant to the construction of the snout in longirostrine pseudosuchians, thalattosuchian salt excretion and regulation, as well as neuroanatomical and sensory adaptations in some of the earliest diverging members of the Crocodylomorpha.

Simplified pneumatic sinuses

Our reconstructions show that the paranasal sinus system in Pelagosaurus and Gavialis (Figs. 3 and 4; dark green) is simplified as compared to other extant crocodylians such as Alligator or bird-line archosaurs (Witmer & Ridgely, 2008), being restricted to the posterior region of the snout (or olfactory region). Simplification of the paranasal sinuses has also been demonstrated in metriorhynchoids (Herrera, Fernández & Gasparini, 2013). Such a morphology is presumably correlated with a longirostrine snout morphology, as longirostrine phytosaurs also have less elaborate paranasal sinuses (Lautenschlager & Butler, 2016). The difference in the extent of the paranasal sinus system in longirostrine versus brevirostrine archosaurs suggests that archosaurian snout development may be influenced by outpocketing of the nasal epithelium (Witmer, 1997) or that the morphology of the skull bones themselves may impose limits on the outpocketing of the nasal epithelium. Irrespective of the underlying developmental mechanism, the simplified paranasal sinuses in longirostrine forms, as demonstrated here in Pelagosaurus and Gavialis, appear to allow the snout to maintain a long, tubular morphology—a construction mechanically beneficial for feeding on fast moving prey (Pierce, Angielczyk & Rayfield, 2008; Pierce, Angielczyk & Rayfield , 2009b). As a longirostrine snout morphology typifies the Thalattosuchia, a simplified paranasal sinus system may characterize the clade, with some exceptions. For instance, the derived metriorhynchoid Dakosaurus andiniensis has a relatively short and tall snout (Gasparini, Pol & Spalletti, 2006), meaning this taxon may have had more elaborate paranasal sinuses. Such a hypothesis should be tested by future work.

Before this study, little was known about the paratympanic sinuses outside of Alligator and bird-line archosaurs (e.g., Witmer & Ridgely, 2008; Dufeau & Witmer, 2015). It is clear from our reconstructions that the system in Gavialis (Fig. 4; dark purple) is similar to Alligator, but that Pelagosaurus (Fig. 3; dark purple) shows a number of differences, some of which are shared with the metriorhynchoid “Metriorhynchus” cf. westermanni (Fernández et al., 2011). The most conspicuous feature of the paratympanic sinus system in Pelagosaurus and Metriorhynchus is that it is confluent with a cavity (Figs. 3B and 3D) that presumably carried the caudal middle cerebral (head) vein (Witmer & Ridgely, 2008; Porter, 2015; Porter, Sedlmayr & Witmer, 2016). As the caudal middle cerebral vein typically drains venous blood from the brain into the internal jugular, it is unclear what the functional implications of this morphology may be (i.e., why such a large volume of venous blood would drain through the paratympanic sinus). Further investigation of this morphology is encouraged, with higher resolution scans and in a broader range of thalattosuchians and crocodylomorphs.

Another key difference is that Pelagosaurus lacks an intertympanic sinus connecting the left and right middle ear cavities above the endocast, as seen Gavialis (Fig. 4; dark purple; also noted by Dufeau, 2011). In modern crocodylians, the tympanic air spaces enhance low-frequency hearing (Witmer & Ridgely, 2008; Dufeau & Witmer, 2015) and the intertympanic space, in particular, helps to conduct vibrations through the head allowing for sound localization (see Dufeau & Witmer, 2015). Thus, the lack of an intertympanic sinus in Pelagosaurus may indicate it was less capable of detecting the direction and distance of sound; although the large cochlea may have compensated for this (see below). Further, the large, laterally placed orbits in thalattosuchians, and particular Pelagosaurus, may indicate these animals were highly visual predators (Pierce & Benton, 2006), relying less on auditory signals. Although the cranial nerves were not distinguishable here, Herrera & Vennari (2015) described enlarged oculomotor nerves (CN III) in the braincase of the metriorhynchid Dakosaurus that may have functioned to control finer-scale eye movements.

A final difference between Pelagosaurus and extant crocodylians is that the ventral aspect of the paratympanic sinus (=Eustachian tube) is not ‘verticalized’ in Pelagosaurus (Figs. 3B and 3D). ‘Verticalization’ of the pharyngeal sinus is possible in modern crocodiles and other eusuchians (Figs. 4B, 4D ) due to the ventral displacement of the basicranium, especially the basioccipital and basisphenoid, due in part to the development of a complete secondary palate (with posteriorly displaced internal choana) and reorganization of the jaw musculature (Tarsitano, Frey & Riess, 1989; Gold, Brochu & Norell, 2014).

Advanced osmoregulation

There is a major difference between the olfactory region of the nasal cavity of Pelagosaurus and that of Gavialis (Figs. 3 and 4; dark green), with the region in Pelagosaurus showing bilaterally symmetrical bulbous expansions of the olfactory recess anterior to the orbits (Figs. 3A, 3C and 3D). We interpret the expansion of the olfactory recesses as an osteological correlate of enlarged nasal salt glands. Such glands have been reported in a similar anatomical position in natural endocasts of various metriorhynchoids (Fernández & Gasparini, 2000; Fernández & Gasparini, 2008; Herrera, Fernández & Gasparini, 2013; Herrera, 2015) and are proposed to have accommodated hypertrophied salt-secretion glands due to their ‘lobular’ surface texture. Previous workers have suggested that enlarged nasal salt glands are a derived feature of the highly-specialized metriorhynchoids, enabling them to maintain constant plasma osmolality (Fernández & Gasparini, 2000; Fernández & Gasparini, 2008). Although Fernández & Gasparini (2008) predicted the presence of salt glands in teleosauroids, they were hypothesized to have been small with low secretory capability. Brusatte et al. (2016) recently hypothesized the presence of a metriorhynchoid-like enlarged salt gland in the Toarcian teleosauroid Steneosaurus based on correlative evidence from the pituitary (see below). Furthermore, Wilberg (2015b) interpreted cavities ventral to the prefrontals and the enlarged carotid foramen/canal in the Middle Jurassic taxon Zoneait as being consistent with the presence of salt glands in an early metriorhynchoid. Here, we document the presence of salt gland osteological correlates directly for an Early Jurassic thalattosuchian, suggesting advanced salt-excretion capabilities were present amongst some of the earliest thalattosuchians.

The most conspicuous feature in the endocast of Pelagosaurus is a pair of greatly enlarged channels extending anterodorsally from the expanded pituitary (Figs. 5B and 5C). A similar channel extending from the pituitary was identified as the optic nerve or CN II in the teleosauroid Teleosaurus eucephalus by Seeley (1880), although CN II is clearly visible further anteriorly, and Wharton (2000) described anteriorly projecting channels from the pituitary region of Steneosaurus as the oculomotor nerve or CN III. However, similar channels were recently described by Brusatte et al. (2016) in a second specimen of Steneosaurus as enlarged orbital arties. Their interpretation proposes that the enlarged internal carotid (also seen in Pelagosaurus and metriorhynchoids; Herrera, 2015; Wilberg, 2015b) and orbital arteries were supplying the salt glands with a large volume of blood for osmoregulation. As cranial nerves are not normally associated with the pituitary, it is most likely that these channels housed arteries that were supplying the salt glands aiding in osmoregulation. Furthermore, the greatly expanded pituitary fossa (=sella turcica) (Figs. 5B and 5C) in Pelagosaurus and Steneosaurus (Brusatte et al., 2016) may have housed an enlarged pituitary gland (=hypophysis). In reptiles, the posterior pituitary is thought to have an antidiuretic effect by constricting glomerular capillaries thereby decreasing blood flow and water loss, a mechanism known as “glomerular antidiuresis” (Heller, 1942; Heller, 1950). This raises the possibility that secretion of an increased volume of antidiuretic hormone (e.g., vasopressin) in thalattosuchians may have also aided in preventing dehydration in a marine environment.

Neuroanatomic adaptations

The endocast of Pelagosaurus shows several characteristics that distinguish it from Gavialis and other crocodile-line archosaurs (Fig. 5). For instance, the cerebrum in Pelagosaurus is relatively large, and is elongated along its anteroposterior length, being symmetrical in form with a deep median longitudinal fissure (Figs. 5A–5C). This contrasts with the smaller cerebral regions of phytosaurs and other crocodylomorphs (with the exception of the notosuchian Simosuchus clarki; Table 2), which are asymmetrical and expanded posteriorly (Figs. 5D–5E; Hopson, 1979; Witmer et al., 2008). A large, symmetrically expanded cerebrum may also be present in the teleosauroids Teleosaurus eucephalus (Seeley, 1880) and Steneosaurus (Wharton, 2000; Brusatte et al., 2016), but the cerebral region in the metriorhynchoid Cricosaurus araucanensis is comparatively small (Table 2; Herrera, Fernández & Gasparini, 2013). In birds (and mammals), larger cerebral regions are associated with refined interpretation of sensory inputs and greater neuronal area to execute increasingly complex behaviors (Rogers, 1999). Although we cannot directly assess behavioral complexity in Pelagosaurus, various other features documented here and in prior literature suggest that Pelagosaurus, and perhaps other thalattosuchians, received greater sensory input from the eyes and labyrinth (see below) than most other pseudosuchians, which is consistent with an enhanced capacity to process sensory information.

Hearing and balance

The morphology of the endosseous labyrinth points towards enhanced sensory capabilities in Pelagosaurus. Compared to Gavialis and other crocodile-line archosaurs, Pelagosaurus has a long endosseous cochlear duct (Figs. 6 and 7; Table 2). Cochlear length has been used as a rough proxy for hearing capabilities in crocodylians and birds (Wever, 1978; Gleich & Manley, 2000; Witmer et al., 2008; Walsh et al., 2009), as it directly relates to the length of the sensory epithelium (or basilar membrane) that stimulates the organ of Corti to transduce mechanical sound vibrations into nerve impulses (Witmer et al., 2008). Thus, the long cochlea in Pelagosaurus suggests an enhanced ability to discriminate auditory stimuli. Brusatte et al. (2016) observed a long cochlear duct in the thalattosuchian Steneosaurus, similar to that of Pelagosaurus. They interpreted this as a plesiomorphic retention of terrestrial-type hearing in a derived marine archosaur lineage. However, short cochlear ducts are not only widespread among pseudosuchians (including Triassic taxa such as phytosaurs; Fig. 7; Table 2) they are also present in early members of the avian stem lineage (the sister taxon of Pseudosuchia) such as Herrerasaurus and Massospondylus (Sereno et al., 2007; Knoll et al., 2012). This suggests that a short, not long, cochlea represents the primitive condition for Pseudosuchia, indicating that the long cochlea seen in Pelagosaurus and Steneosaurus is a derived morphology (Fig. 7). Furthermore, there is no evidence that adaptation to aquatic life in tetrapods involves reduction of the cochlea. For example, the relationship between cochlea length and body mass in cetaceans is similar to that in terrestrial mammals (Spoor et al., 2002). Marine tetrapod lineages that evolved from terrestrial ancestors with impedance-matching middle ears, including mosasauroids, sea turtles, cetaceans, pinnipeds and other taxa have retained use of the tympanic route in underwater sound perception, which may result in improved ability to localize the direction of the sound sources (Hetherington, 2008). The key modifications to auditory anatomy seen in secondarily aquatic taxa involve features of the middle ear, especially the stiffness and size of the tympanum (Hetherington, 2008). Information of such characteristics are currently unavailable for thalattosuchians, so the question of whether they had ‘terrestrially adapted’ hearing remains open.

In addition to hearing, the morphology of the vestibular system—the sensory organ of balance—is unlike that seen in Gavialis and other pseudosuchians, with a few exceptions among crocodylians (e.g., Crocodylus johnstoni; Fig. 7). In Pelagosaurus, the anterior and posterior canals form a ‘pyramidal’ shape and the posterior canal is relatively large, especially compared to modern crocodylians (Fig. 6). A similar vestibular shape can be seen in the teleosauroid Steneosaurus (Brusatte et al., 2016), indicating that this morphology may be more widespread among thalattosuchians. Semicircular canals sense angular rotations of the head and increases in their size have been linked to enhanced agility and aerobatic ability (Witmer et al., 2003; Alonso et al., 2004; Spoor et al., 2007; Ekdale, 2015). Distinct labyrinth morphologies are present in the most aquatic members of many extant tetrapods (Georgi & Sipla, 2008; Spoor & Thewissen, 2008), including cetaceans (Spoor et al., 2002) and carnivoran mammals (Gröhe et al., 2016), although they do not seem to be present in diving birds (Smith & Clarke, 2012). Furthermore, distinct morphological adaptations to aquatic life are seen in the labyrinths of different groups among reptiles and mammals (Georgi & Sipla, 2008; Spoor & Thewissen, 2008; Yi & Norell, 2015), so general patterns of change in labyrinth morphology in response to the evolution of aquatic life seem to be absent. Nevertheless, the distinct labyrinth morphology of Pelagosaurus (and Steneosaurus) is most likely explained by its distinct locomotor ecology—representing an early member of the most aquatically-adapted clade of pseudosuchians. This hypothesis will be tested as additional data on pseudosuchian, and particularly thalattosuchian, labyrinths become available, and potentially also by comparative studies of crocodylomorph labyrinth evolution.

Conclusions

Based on our analysis of the endocranial anatomy of Pelagosaurus typus and Gavialis gangeticus, in comparison to other thalattosuchians and pseudosuchian taxa, we propose the following thalattosuchian characteristics: (1) a pyramidal morphology of the semicircular canals; (2) an elongate endosseous cochlear duct (indicating greater sensitivity to hearing); (3) large, paired channels extending anteriorly from an enlarged pituitary fossa that may have housed the orbital artery; (4) a relatively straight brain (possibly due to the presence of large, laterally placed orbits); and (5) an enlarged venous sinus projecting dorsally from the endocast which is confluent with the paratympanic sinus system. Further to this, we found that Pelagosaurus possessed a large bulbous expansion of the nasal cavity anterior to the orbits, homologous in structure to that which houses a hypothesized salt gland in Late Jurassic metriorhynchoids, providing evidence that this physiological adaptation evolved early in thalattosuchian evolution. Finally, the pyramidal semicircular canals of thalattosuchians, long cochlear duct, enlarged pituitary fossa and early evolution of a hypothesized salt gland may reflect a high level of sensory and physiological adaptation to aquatic life in this clade, occurring well in advance of postcranial adaptations to marine open water swimming.

Supplemental Information

Figure S1 Interactive three-dimensional reconstruction of the skull and endocranial morphology of Pelagosaurus typus (BRLSI M1413)

Note, best viewed with CAD Optimized Lighting.

Click here for additional data file.

Figure S2 Interactive three-dimensional reconstruction of the skull and endocranial morphology of Gavialis gangeticus (UMZC R 5792)

Note, best viewed with CAD Optimized Lighting.

Click here for additional data file.

We thank Matthew Williams at the BRLSI and Matthew Lowe at the UMZC for loaning material used in this study; Paul Gignac and Nathan Kley for anatomical identification assistance; John Hutchinson for helping to transport the gharial skull to and from The Royal Veterinary College; Matthew Colbert and the University of Texas High-Resolution X-ray CT Facility for scanning the fossil material; and Timothy Rowe at the University of Texas at Austin who provided funding for scanning through the National Science Foundation Digital Libraries Initiative. Further, we thank Ariana Paulina Carabajal and Yanina Herrera for providing thorough and constructive comments that greatly improved the quality of the paper. Megan Williams submitted a version of this study as a Part II thesis in the Department of Zoology, University of Cambridge.

Additional Information and Declarations

Competing Interests

Author Contributions

Data Availability

Stephanie E Pierce is an Academic Editor for PeerJ.

Stephanie E Pierce conceived and designed the experiments, performed the experiments, analyzed the data, contributed reagents/materials/analysis tools, wrote the paper, prepared figures and/or tables, reviewed drafts of the paper.

Megan Williams performed the experiments, analyzed the data, wrote the paper, reviewed drafts of the paper.

Roger B.J. Benson analyzed the data, wrote the paper, prepared figures and/or tables, reviewed drafts of the paper.

The following information was supplied regarding data availability:

Interactive 3D PDFs of the anatomical reconstructions are provided as supplementary figures to this paper. In addition, the CT data for Gavialis gangeticus has been reposited in the University Museum of Zoology, Cambridge, and the CT data for Pelagosaurus typus is stored on DigiMorph (http://digimorph.org/specimens/Pelagosaurus_typus/whole/).

Once the paper is accepted for publication, we will make the official inquiries to submit our 3D object files to MorphoMuseum; although the final decision is up to the museums that hold copyright to the specimens.

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
