# Peer review of "Virtual reconstruction of the endocranial anatomy of the early Jurassic marine crocodylomorph Pelagosaurus typus (Thalattosuchia)"

_PeerJ, doi:10.7717/peerj.3225_

## Round 0.1 · original submission · Major Revisions

I now have two reviews back for your submitted manuscript titled "Virtual reconstruction of the brain and sinuses of the early Jurassic marine crocodylomorph Pelagosaurus typus (Thalattosuchia)". The reviewers raised some serious concerns. Therefore, my decision is "Major Revision". With the revised version of your manuscript, I invite you to provide a "Response to referees" document in which you list all suggestions and corrections, together with your response to each. I will then send the new version out again for review to the two reviewers.

·

Basic reporting

The authors present here the most complete study on the neuroanatomy (brain and inner ear), nasal cavity and pneumaticity of an extinct marine reptile made so far. The work is original and the description is easy to follow. However, see my comments about the terminology used in paleoneurology, including the title, descriptions, and Tables 1 and 2 (comments are indicated in the pdf).
I did-not make any grammar corrections since English is not my native language.

Experimental design

No comments

Validity of the findings

No comments

Additional comments

The authors present here the most complete study on the neuroanatomy (brain and inner ear), nasal cavity and pneumaticity of an extinct marine reptile made so far. The work is original and the description is mostly clear (see my comments about the terminology used in paleoneurology, including the title, descriptions and Tables 1 and 2), with a complete background and a complete bibliography. The knowledge of thalattosuchian paleoneurology is poor and based in few and incomplete specimens, reason why this complete skull is so informative and will be used in the future to better understand other less complete specimens. In first place, this paper adds novel anatomical information on the above mentioned structures for the taxon in particular, but also for the clade. Endocranial morphology is not often used in diagnoses and even less treated in character matrixes. Therefore, the anatomical information presented here has a potential use in phylogeny. In addition, the authors present digital reconstructions of the brain, inner ear and nasal cavities of a living Gavialis. In second place, the morphology of the brain, inner ear, nasal cavities and skull pneumaticity is interpreted by the authors under the light of adaptation and evolutionary trend. This is important for the analysis of the paleobiological implications of several neuroanatomical structures, which are poorly understand, and will be of interest for other researchers worldwide. In fact, I waited years to actually see a complete thalattosuchian cranial endocast and inner ear, and compare it with the terrestrial reptiles I work with. Congratulations to the authors!.
-Methods (regarding CT scannings, production of images and storage of digital data) are described with sufficient detail
-the illustrations are fantastic (they are high quality, easy to read, and well designed, and show all the structures mentioned in the text).

·

Basic reporting

Some useful literatura is missing.
Figures 3 and 4. I think the authors should add labels on both figures. This would be very helpful to understand the interpretations described.
For example: lines 235-237: "In Pelagosaurus and Gavialis, there is a subconical, subsidiary outpocketing ...... to the nasal passage (Figs. 3D, E, 4D, E; dark green)"), here, the subsidiary outpocketing should be label.
Also, "The internal naris of Gavialis (Fig. 4D, E; dark yellow) is enclosed by the pterygoids..". The authors should label the internal naris in figure 4.

Experimental design

No Comments

Validity of the findings

See in pdf attached the comments about conclusions

Additional comments

The manuscript "Virtual reconstruction of the brain and sinuses of the early Jurassic marine crocodylomorph Pelagosaurus typus (Thalattosuchia)" will be an important contribution to the growing body of knowledge about the soft anatomy and evolution of thalattosuchians. The paper presents the first description of the internal structures of Pelagosaurus typus, a thalattosuchian with a controversial phylogentenic position, thus it is useful for reptile specialist and non-specialist as it brings innovative information about soft anatomy of crocodyliform that can be useful for paleobiological and systematic studies.
While I think the manuscript should be published, I believe it requires major revisions.

General comments

The authors used the paper of Herrera, Fernández and Gasparini (2013) to take some measurements (expressed on Tables 1 and 2). In this paper, we did not describe the brain (just the olfactory tract and bulbs), so the 3D reconstruction is not accurate. I really suggest be careful with the measurements and the interpretations that you took from my 3D models.

The description of narial cavity and associated structures is a little bit general. For example, the authors describe the narial cavity referring to the "dark yellow" structure that they reconstruct, but they did not distinguish between the different parts that form the nasal cavity (e.g. nasal cavity proper and nasopharyngeal ducts) and that are recognizable in Figure 3. Also, the primary choana is identifiable and the authors did not mention this structure. The use of terms like dark yellow or dark green to describe some structures is not specific. I believe that the description and related figures should be improved.

Although the authors say that the olfactory region is used here in a loose sense, I think that use "olfactory recesses" to do reference to the osteological correlates of the salt gland is not proper and is confusing, I strongly suggest the use of other term (e.g. antorbital recesses, prefrontal recesses). Also, I suggest replacing narial cavity by nasal cavity, narial passage by nasal cavity proper or nasopharyngeal duct when appropriate (e.g. line 210. narial passage should be replaced by nasopharyngeal duct ).

Related to the antorbital fenestra, the authors go for the classical point of view (i.e. Pelagosaurus typus has antorbital fenestra). However, Witmer (1997), and Jouve (2009) proposed that basal thalattosuchians have an internalized antorbital fenestra. The authors did not discuss about this interpretation. Based on the description and figure 3, my opinion is also that in Pelagosaurus typus the antorbital fenestra is internalized. The morphology and the topographic relationships of the structures in the dark green region resemble the morphology present in Cricosaurus araucanensis (i.e., the antorbital sinus is ventral to the salt gland, and tapers anteriorly along the length of the snout, the antorbital sinus is lateral to the primary choana and, apparently, the ostium is directly opposite to the it), which has an internalized antorbital fenestra. I am not able to distinguish/understand the participation of the antorbital sinus to the osteological correlate of the salt gland, as the authors described. But, if so, I believe that is not enough evidence (or it is not properly described) to assume the presence of antorbital fenestra because there are more evidences to hypothetized the absence of external antorbital fenestra.
Leardi et al. (2012) proposed, based on a dynamic homology approach, that the fenestra in basal thalattosuchians is interpreted as homologous to the antorbital fenestra of other archosaurs. As noted, there are many sources of information related to this topic that should be discussed as I think that it is an important topic for understand the evolution of thalattosuchians and I feel that the authors did not take the opportunity to describe and discuss it properly.

There are some missing literature that I think have to be included, like:
Dufeau (2011) in his doctoral dissertation (The Evolution of Cranial Pneumaticity in Archosauria: Patterns of Paratympanic Sinus Development) identified the sinuses that form the paratympanic sinus system in Pelagosaurus typus. He did not describe them deeply but there are some comments and a figure that can be used to comparisons.
The authors used the names, sinus 1, 2 or 3 to referred to the sinuses, but they do not mention/suggest the homology of them.
Wilberg (2015) in his paper: "A new metriorhynchoid (Crocodylomorpha, Thalattosuchia) from the Middle Jurassic of Oregon and the evolutionary timing of marine adaptations in thalattosuchian crocodylomorphs", suggested the presence of a salt gland in Zoneait (a metriorhynchoid), described a greatly enlarged carotid foramen and a large internal carotid canal as well as an enlarged dorsal sinus. This taxon is not included in this manuscript, and to trace some features in a evolutionary context I believe that it has to be included because it is the only known metriorhynchoid non metriorhynchid with some available information about soft tissues.

The authors suggest a relation between the lateralization of the orbits and the reduction of the flexures of the brain. However, similar values are given for Cricosaurus araucanensis (CF=155º, PF=155º) and Gavialis gangeticus (CF=150º, PF=154). In thalattosuchians, the metriorhynchids (like C. araucanensis) have strictly lateral orbits, on the other hand Gavialis gangeticus does not have lateral orbits. So, I'm thinking that this interpretation is wrong or the measurements of C. araucanesis (see above) are wrong.



additional comments are included in the pdf file

---

## Round 0.2 · Minor Revisions

Thank you very much for submitting the revised version of your manuscript. I am quite positive, but I need you to address the few suggestions made by Reviewer #2 prior to formal acceptance.

·

Basic reporting

The structure of the article is fine and is clear. Missing literature was added.

Experimental design

no comment

Validity of the findings

no comment

Additional comments

I reviewed a previous version of the paper that has been improved, as I said before this work is an important contribution to the growing body of knowledge about the soft anatomy and evolution of thalattosuchians.
I have only few comments, which are highlighted in the pdf.

---

## Round 0.3 · accepted · Accept

Dear Stephanie,

I am pleased to inform you that your manuscript is now accepted for publication in PeerJ.

Please, don't forget to make all the raw data available in a permanent public repository.

Thank you for your contribution!

Best regards,
Fabien